# In Vitro and In Vivo Activity of AS101 against Carbapenem-Resistant *Acinetobacter baumannii*

**DOI:** 10.3390/ph14080823

**Published:** 2021-08-21

**Authors:** Tsung-Ying Yang, Sung-Pin Tseng, Heather Nokulunga Dlamini, Po-Liang Lu, Lin Lin, Liang-Chun Wang, Wei-Chun Hung

**Affiliations:** 1Department of Medical Laboratory Science and Biotechnology, College of Health Sciences, Kaohsiung Medical University, Kaohsiung 807, Taiwan; zegma040899@gmail.com (T.-Y.Y.); tsengsp@kmu.edu.tw (S.-P.T.); heather.dlamini@gmail.com (H.N.D.); 2Department of Marine Biotechnology and Resources, National Sun Yat-sen University, Kaohsiung 80424, Taiwan; marknjoy@g-mail.nsysu.edu.tw; 3Drug Development and Value Creation Research Center, Kaohsiung Medical University, Kaohsiung 807, Taiwan; 4Graduate Institute of Animal Vaccine Technology, College of Veterinary Medicine, National Pingtung University of Science and Technology, Pingtung 912, Taiwan; 5Center for Liquid Biopsy and Cohort Research, Kaohsiung Medical University, Kaohsiung 807, Taiwan; d830166@gmail.com; 6Department of Internal Medicine, Division of Infectious Diseases, Kaohsiung Medical University Hospital, Kaohsiung 807, Taiwan; 7School of Post-Baccalaureate Medicine, College of Medicine, Kaohsiung Medical University, Kaohsiung 807, Taiwan; 8Department of Culinary Art, I-Shou University, Kaohsiung 84001, Taiwan; lynnlin@isu.edu.tw; 9Department of Microbiology and Immunology, College of Medicine, Kaohsiung Medical University, Kaohsiung 807, Taiwan

**Keywords:** drug repurposing, AS101, carbapenem resistance, *Acinetobacter baumannii*

## Abstract

The increasing trend of carbapenem-resistant *Acinetobacter baumannii* (CRAB) worldwide has become a concern, limiting therapeutic alternatives and increasing morbidity and mortality rates. The immunomodulation agent ammonium trichloro (dioxoethylene-O,O′-) tellurate (AS101) was repurposed as an antimicrobial agent against CRAB. Between 2016 and 2018, 27 CRAB clinical isolates were collected in Taiwan. The in vitro antibacterial activities of AS101 were evaluated using broth microdilution, time-kill assay, reactive oxygen species (ROS) detection and electron microscopy. In vivo effectiveness was assessed using a sepsis mouse infection model. The MIC range of AS101 for 27 CRAB isolates was from 0.5 to 32 µg/mL, which is below its 50% cytotoxicity (approximately 150 µg/mL). Bactericidal activity was confirmed using a time-kill assay. The antibacterial mechanism of AS101 was the accumulation of the ROS and the disruption of the cell membrane, which, in turn, results in cell death. The carbapenemase-producing *A. baumannii* mouse sepsis model showed that AS101 was a better therapeutic effect than colistin. The mice survival rate after 120 h was 33% (4/12) in the colistin-treated group and 58% (7/12) in the high-dose AS101 (3.33 mg/kg/day) group. Furthermore, high-dose AS101 significantly decreased bacterial population in the liver, kidney and spleen (all *p* < 0.001). These findings support the concept that AS101 is an ideal candidate for further testing in future studies.

## 1. Introduction

*Acinetobacter baumannii* is a causative organism of nosocomial infections, including pneumonia, sepsis and urinary tract infection, and its propensity to acquire multidrug, extensively drug and pan-drug resistance make it a global threat in healthcare settings [1,2]. Carbapenems were considered as one of the last-line resorts against *A. baumannii* infections. However, due to the increasing consumption of carbapenems, the increasing trend of carbapenem resistance in *A. baumannii* (CRAB) worldwide has become a concern, limiting therapeutic alternatives and increasing morbidity and mortality rates. In a previous study, Jamie et al. reviewed published reports from nosocomial settings in Latin America published between 2002 and 2013 [3]. Rates of carbapenem resistance were found as high as 90% for *A. baumannii* isolates across Latin America. In a surveillance study in the US, 1490 *A. baumannii* isolates were collected from healthcare-associated infections during 2009 to 2010, and 80.6% (1201/1490) of the isolates were resistant to carbapenems [4]. In another surveillance study from Europe in 2019, 6113 isolates of the *Acinetobacter* species were reported from 30 countries, and among them, 5953 isolates possessed antimicrobial susceptibility testing (AST) results for carbapenems [5]. The overall carbapenem-resistant rate in Europe was 32.6%, whereas the resistant rates in some countries were higher than 50%, particularly in East and Southern Europe.

In this scenario, new drugs are urgently needed, whereas drug development is a time-consuming, high-investment and high-risk process [6,7]. Thus, drug repurposing is considered a much more efficient approach as it is economical and riskless in the drug discovery phase. Anticancer drugs are promising as antimicrobials due to the sample similarities between growing tumors and bacterial infections such as the high replication rates, high dissemination tendency, high immune system resistance and the tendency to become insensitive to treatments [8]. Brian et al. discovered that an anticancer drug, mitomycin C (MMC), has remarkable antibacterial properties against several bacterial pathogens, which include *Escherichia coli*, *Staphylococcus aureus* and *Pseudomonas aeruginosa* [9]. Martha et al. conducted a study where they evaluated the antibacterial effects of MMC against multiple clinical strains of MDR *A. baumannii* and increase the survival of *Galleria mellonella* that was infected with a lethal dose of each strain from 0 to 53–80% [8]. 

AS101 was an immunomodulator that can induce human lymphocytes to proliferate cytokines such as IL-2 and CSF (colony-stimulating factor) secretion [10,11]. An immune-directed treatment effect of AS101 was also observed in a cecal ligation-and-punctured mouse model in a previous study [12]. The low toxicity of AS101 including a 50% cytotoxic concentration of AS101 in Vero cells (145 μg/mL) and a 50% lethal dose (LD_50_) for intravenous injections in mice (10 mg/kg) indicated the potential for clinical use [13,14]. The clinical trial phase I of AS101 for HIV infections and phase I/II studies for external genital warts were completed in 2012 and 2013 (NCT00001006 and NCT01555112), respectively, and it is currently undergoing phase I/II studies in patients with aging macular degeneration (NCT03216538). Research in various fields has found that AS101 has a good effect on a variety of diseases, including clinical conditions involving immunosuppression, malignant tumors and AIDS, autoimmune diseases, antiviral, antiparasitic and other diseases [13,15,16,17]. The antibacterial effects of AS101 were also revealed against ESBL-producing *Klebsiella pneumoniae* and *Enterobacter cloacae* [18,19]. Recently, another examination of AS101 against colistin and carbapenem-resistant *K. pneumoniae* also documented the antibacterial potential of AS101 [20]. According to these contributions, we aimed to re-evaluate AS101 as a novel antimicrobial agent against carbapenem-resistant *A. baumannii* by in vitro antibacterial activities, antibacterial mechanisms and in vivo effectiveness in the sepsis mouse infection model.

## 2. Results

### 2.1. Antibacterial Efficacy of AS101 against CRAB Clinical Isolates

Among these 27 CRAB clinical isolates, high resistant ratios were found in the following different classes of antibiotics: meropenem (27/27, 100%), levofloxacin (26/27, 96.3%), ticarcillin (27/27, 100%), ceftazidime (27/27, 100%), doxycycline (25/27, 92.6%) and gentamicin (22/27, 81.5%). No resistance was found in colistin (0/27, 0%) (Table 1). Carbapenemase were found in 8 isolates with *bla*_OXA-24_, 23 with *bla*_OXA-69_, 1 with *bla*_VIM-2_ and 1 with *bla*_IMP-1_, respectively. AS101 showed antibacterial effect in all the clinical strains, even though these strains harbored carbapenemase genes. The MIC range was from 0.5–32 μg/mL, and 90% of the strains were inhibited by a concentration as low as 8 μg/mL (Table 1). The MIC values showed are significantly lower than the 50% cytotoxic concentration of AS101 in the Vero cell (145 μg/mL), suggesting low cytotoxicity at effective concentrations [13].

### 2.2. Characterization of Antibacterial Activity of AS101

A time-kill assay was performed in a CRAB TSP-AB-03 strain that harbored the *bla*_OXA-69_ and *bla*_OXA-24_ genes. In order to understand the antibacterial activity of AS101, minocycline was selected as the bacteriostatic control agent and rifampin as the bactericidal control agent (Figure 1a,b). After 8 h, AS101 at 1×, 2× and 4× MIC eliminated more than 99.9% of the bacteria (Figure 1c), showing that AS101 acts as a bactericidal agent against carbapenemase-producing *A. baumannii.*

### 2.3. Antibacterial Mechanism of AS101

To understand antibacterial mechanism of AS101, the combination tests of chemical agents with distinct functions and AS101 were determined. Table 2 showed that the MIC values of AS101 were significantly decreased when co-cultured with 1 mM EDTA, from 2 to 0.21 μg/mL, indicating that the antibacterial activity of AS101 was strongly impacted by the outer-membrane permeability and the antibacterial mechanism of action might be inside the outer-membrane. The MIC values of AS101 were not significant different between the co-cultured with 10 mM calcium or magnesium ions and the normal group. The drastic MIC change observed in AS101 is co-cultured with 500 mM Mannitol, increasing from 2 to 24 μg/mL. As an ROS scavenger, mannitol clears the reactive oxygen species (ROS), leading to the inability of AS101. Taken together, it can be inferred that AS101 enters the bacterial outer membrane and causes the accumulation of the ROS, eventually leading to the death of the bacteria. As suggested by the results above, the bactericidal ability of AS101 may be related to the generation of ROS. DCFH-DA was used to detect the generation of ROS in CRAB TSP-AB-03 incubated with different concentrations of AS101. Compared with the control group, the 1× MIC (*p* < 0.01), 2× MIC (*p* < 0.01) and 4× MIC (*p* < 0.01) groups showed a statistically significant and dose-dependent increasing trend of ROS generation in CRAB TSP-AB-03 (Figure 2). These results confirmed that ROS generation is related to the antibacterial activity of AS101.

### 2.4. Effect on Bacterial Morphology

The TSP-AB-03 strain was treated with 1× MIC (2 μg/mL) of AS101 and its surface morphological changes, compared to the untreated group (control), were observed using a scanning electron microscope (SEM) (Figure 3). Figure 3a,b showed the morphological structure of the untreated group was intact and without any damages at 5000× and 10,000× magnification. The morphology of the AS101-treated bacteria was greatly changed (Figure 3c,d), with cell sinking in the middle that resembles a disruption in the cell membrane integrity. Compared to the untreated group (Figure 4a,b), transmission electron microscope (TEM) showed that ghost cells were the most evident structural changes found in AS101-treated bacteria at 12,000× and 20,000× magnification (Figure 4c,d), with also the distortion of the cell sizes and shape in the micrograph. These SEM and TEM results suggest that AS101 could lead to cell lysis and the disruption of the cell membrane, which, in turn, results in cell death.

### 2.5. In Vivo Administration Test

The therapeutic effect of AS101 on the TSP-AB-03 strain was tested using a mouse sepsis model that was intraperitoneally injected with a lethally dose of bacteria (10^8^ CFU in 0.1 mL). The placebo group, 17% (1/6) of the mice, survived and the rest died within 30 h post-infection (Figure 5a). In the negative control (meropenem, 200 mg/kg/day, QID) group, the same trend was observed; in the positive control (colistin methanesulfonate (CMS), 40 mg/kg/day, QID) group, the results were slightly improved with 33% (4/12) mice surviving 120 h post-infection; with the low-dose AS101 (1.67 mg/kg/day), the mice survival rate after 120 h was 42% (5/12); a great improvement was observed with the high-dose AS101 (3.33 mg/kg/day), where 58% (7/12) of mice survived after 120 h. These results show that AS101 has a better therapeutic effect on mice infected with carbapenemase-producing *A. baumannii*, compared to the currently in clinical use colistin. 

The mice were sacrificed 18 h post infection, and the liver, kidney and spleen were collected for the quantification of the bacterial load on organs (Figure 5b–d). Compared to the placebo group, after treatment with high-dose AS101 (3.33 mg/kg), the mice showed a significant reduction in liver bacterial load (*p* < 0.001), while there was no significant difference with either CMS or low-dose AS101 (1.67 mg/kg) (Figure 5b). Figure 5c showed a significant reduction in the kidney bacterial load in the groups with low-dose (*p* < 0.05) and high-dose (*p* < 0.01) AS101, while no significant difference was noticed in the group with CMS. Compared to the placebo group, after treatment with high-dose AS101 (3.33 mg/kg), the mice showed a significant reduction in the spleen bacterial load (*p* < 0.001), while there was no significant difference with either CMS or low-dose AS101 (1.67 mg/kg) (Figure 5d).

## 3. Discussion

Carbapenem-resistant *A. baumannii* (CRAB) have become a worldwide problem and defined as critical priority pathogens for the development of new antibiotics by the WHO in 2017 [21]. The carbapenem resistance rate increase yearly and the mortality rates for the most common infections of CRAB approach 60% [22]; in hospitals in London and Southeast England, 30% of samples collected in over 2 years were found to be CRAB [23]; in Latin America, carbapenem resistance rates were found to be as high as 90% [3]; and in Australia, Anton et al. recovered 90 *A. baumannii* isolates from blood specimens from 69 patients, and 58 (64%) were resistant to meropenem [24]. Taiwan has experienced a rapid increase in *A. baumannii* complex-related infections since 1990 and the carbapenem-resistance rate increased from 3.4% in 2002 to 58.7% in 2010. The Taiwan Nosocomial Infections Surveillance (TNIS) data showed an increase in the percentage of CRAB complex infections in the ICU starting from <20% in 2003 to 70% in 2011 [25]. 

Drug repurposing has been regarded as an alternative strategy for drug development. Kimberly et al. used a substructure-based literature search that led to the identification of ciclopirox, an off-patent, topical antifungal drug that was developed approximately forty years ago [26]. It is active against multidrug resistant *E. coli, K. pneumoniae* and *A. baumannii* with the following MIC ranges: 5–15 μg/mL, 5–15 μg/mL and 5–7 μg/mL, respectively. In this study, a synthetic organo-tellurium immunomodulator, AS101, was selected for drug repurposing, owing to its variety of potential therapeutic applications. The MIC range of AS101 against CRAB was from 0.5–32 μg/mL, and 90% of the strains were inhibited by a concentration as low as 8 μg/mL (Table 1). These concentrations were significantly lower than its cytotoxicity concentration (approximately 150 μg/mL), indicating AS101 can be considered as a highly potential antibacterial agent [13]. The time-kill assay showed that AS101 acts as a bactericidal agent against carbapenemase-producing *A. baumannii* and has a stronger antibacterial activity as the concentration increases (Figure 1). 

A previous study showed that divalent cations are necessary to maintain the stability of the outer membrane [27]. EDTA can be used to chelate these cations, thereby resulting in a hyperpermeable bacterial membrane. Our result showed that the MIC values of AS101 were significantly decreased when co-cultured with 1 mM EDTA, indicating that a compromised outer-membrane permeability brings rise to the speculation that the antibacterial mechanism of action might be inside the bacterial cell (Table 2). To further evaluate the mode of action for this drug, its change in MIC value after the addition of the hydroxyl radical scavenger [28], mannitol, was evaluated. Our results showed a 12-fold increase in the MIC of AS101 after the removal of the hydroxyl radicals by mannitol treatment. The increased ROS level also confirmed that the antibacterial activity of AS101 was related to the accumulation of ROS (Figure 2). Previous studies documented that the mode of action for bactericidal antibiotics (β-lactams, quinolones, aminoglycosides), in addition to their main targets, is also related to the cell death mediated by ROS [29,30,31]. These data indicated that AS101 would enter the bacterial outer membrane and cause effects on the bacterial metabolism, leading to the accumulation of the ROS, and kill bacteria.

Bacterial morphology was observed using SEM and TEM (Figure 3 and Figure 4). The cell sinks deeper in the middle, which resembles a disruption in the cell membrane integrity from SEM. This finding was similar to a previous study that evaluated the efficacy of antibiotic treatment on *Clostridium difficile* [32]. In the TEM, ghost cells are the most evident structural changes observed, there is also a distortion of the cell sizes and shapes. Ghost cells are anucleate cells indicative of bacterial cell death [33]. These results suggest that AS101 causes cell-lysis to the bacteria, as indicated by the presence of ghost cells and also disrupts the cell membrane, which, in turn, results in cell death.

Although colistin was one treatment choice against CRAB, treatment efficacy was poor in these strains. One study evaluated the efficacy of intratracheal CMS, imipenem and meropenem in BALB/c mice with carbapenem-resistant *A. baumannii* pneumonia [34], CMS only showed a 33% mice survival rate at 72 h after inoculation and no significant decrease in the pulmonary bacterial loads from 24 to 72 h after treatment. It could be assumed to be as a result of poor tissue penetration and also a slow transformation of CMS into colistin sulfate in the bronchoalveolar lavage fluid, yet the transformation is a prerequisite for the antibacterial activity of CMS [35]. Our mouse sepsis model showed that AS101 was a better therapeutic effect than the currently clinically used colistin (Figure 5a), and the organ load results showed similar data (Figure 5b–d). Our in vivo results go on to further show that compared with CMS, AS101 is more effective in reducing the organ bacterial load and, thereby, recues the mice infected with the carbapenem-resistant *A. baumannii* strain. Furthermore, the effective antibacterial dose of AS101 (1.67 and 3.33 mg/kg) was lower than its 50% lethal dose (LD_50_) (10 mg/kg), indicating the potential for clinical use [13,14]. These findings support the concept that AS101 is an ideal candidate for further testing in future studies.

## 4. Materials and Methods

### 4.1. Bacterial Isolates

Between 2016 and 2018, 27 CRAB clinical isolates were obtained from blood specimen of patients from Kaohsiung Medical University Hospital.

### 4.2. Antimicrobial Susceptibility Testing

The minimum inhibitory concentration (MIC) of antibiotics were determined using the standard agar dilution method. Antibiotics included meropenem, colistin, gentamicin, doxycycline, ticarcillin, levofloxacin and ceftazidime. Two quality-control strains, *Escherichia coli* ATCC 25922 and *Pseudomonas aeruginosa* ATCC 27853, were employed to validate the result of each testing. The interpretations of susceptibility results were according to guidelines established by the Clinical and Laboratory Standards Institute (CLSI) [36].

### 4.3. Antibacterial Activity of AS101

MIC of AS101 was determined using the broth microdilution method. Briefly, AS101 (Development Center for Biotechnology, Taipei, Taiwan) was dissolved in 99% ethanol (5% final concentration) and diluted with brain–heart infusion (BHI) broth, followed by serial 2-fold dilution to 0.5–128 μg/mL per well. Final bacterial concentration was 5 × 10^5^ CFU/mL. The OD_600_ nm absorbance value was measured using a microplate reader at 0 and 18 h after being cultured in a 37 °C incubator. Finally, the MIC value was calculated based on the change in absorbance.

### 4.4. Detection of Carbapenemase

Plasmid DNA was prepared using Presto^TM^ Mini Plasmid Kit (PDH300) (Geneaid, Taipei, Taiwan). Genomic DNA was purified manually as described in a previous study [37]. PCR methods were used to detect carbapenemase including *bla*_KPC_, *bla*_NDM_, *bla*_IMP-1_, *bla*_IMP-2_, *bla*_NMC_, *bla*_SME_, *bla*_VIM-1_, *bla*_VIM-2_, *bla*_SPM-1_, *bla*_GIM-1_, *bla*_SIM-1_, *bla*_IMI_, *bla*_GES_, *bla*_OXA-24_, *bla*_OXA-48_ and *bla*_OXA-69_, respectively [38,39]. The positive PCR products were subject to sequencing for validation of the results. Positive controls were also included in each reaction to rule out operation error.

### 4.5. Time-Kill Kinetics Assay

Time-kill assays were performed as in a previous study [40]. Briefly, bacterial strain TSP-AB-03 was adjusted to 10^6^ CFU/mL in BHI broth and incubated with 1×, 2× and 4× MIC of AS101, and 5% ethanol (control) at 37 °C. Bacterial population was measured at different time-intervals (0, 2, 4, 8 and 24 h). Serial 10-fold dilutions were generated in 1× PBS and plated on LB agar. Plates were then incubated in 37 °C for 18 h and the number of colonies were counted on each plate with the counting interval between 25–250 colonies. Rifampin and minocycline were used as bactericidal and bacteriostatic agent controls, respectively. 

### 4.6. Pharmacological Manipulation

MIC values of AS101 with varies chemical agents were determined using the broth microdilution method according to the aforementioned methods [41]. Thereafter, the chemical agents (EDTA, Ca^2+^, Mg^2+^ and Mannitol) were mixed, individually, with different concentrations of AS101 to observe any changes in MIC values. Into a 96-microwell plate, 100 μL of each solution was added (in their respective concentrations). Then, 100 μL of adjusted bacterial solution was added into each well and the absorbance value was read at 0 and 18 h after incubation at 37 °C.

### 4.7. Reactive Oxygen Species (ROS) Generation Test

Intrabacterial ROS were detected using a 2′, 7′-dichlorodihydrofluorescein diacetate (DCFH-DA) agent as in our previous study [42]. Briefly, bacterial strain TSP-AB-03 was adjusted to McFarland 0.5 in BHI broth and incubated with DCFH-DA (obtaining a final DCFH-DA concentration of 100 μM) for 2 h at 37 °C. DCFH-DA-treated cells were collected, washed with 1× PBS and resuspended to 10^6^ CFU/mL in BHI broth. Bacterial suspension incubated with 1×, 2× and 4× MIC of AS101, and 5% ethanol (control) for 6 h at 37 °C. Fluorescence intensities were detected with a spectrofluorometric reader at 500 and 530 nm wavelengths for excitation and emission, respectively. For the normalization, the change in the fluorescence value between 0 and 6 h was divided by the viable bacterial number. All experiments were performed in triplicate.

### 4.8. Electron Microscopy

Electron microscopy was used to observe contribution of AS101 on bacterial morphology. Bacterial cells were treated with AS101 at 2 μg/mL MIC for bacterial strain TSP-AB-03 for 1 h prior to collection. Scanning electron microscopy procedures were performed, as in our previous study [42]. Sample preparation of TEM was the same condition of SEM and then fixed, stained and prepared following the transmission electron microscopy procedures, as in a previous study [43].

### 4.9. Animal Model

Six-week-old ICR (CD1) male mice were purchased from Lasco Biotech and held in the Kaohsiung Medical University (KMU) Laboratory Animal Center for 1 week prior to use in all experiments. Procedures were submitted for approval by the KMU Institutional Animal Care and Use Committee (No. 106191), and all animal experiments were conducted in accordance with the animal institutional guidelines. The animal study was conducted in the KMU Laboratory Animal Center, an Association for Assessment and Accreditation of Laboratory Animal Care International (AAALAC)-accredited facility.

### 4.10. Bacterial Infection and Survival

The sepsis mouse infection model was established according to previous studies [44,45]. Briefly, mice were injected intraperitoneally with lethal doses of CRAB TSP-AB-03 isolates (10^8^ CFU) and administrated intraperitoneally with AS101, meropenem, colistin (colistin methanesulfonate, CMS), or a PBS vehicle 30 min post-infection. Daily doses of AS101 were 1.67 and 3.33 mg/kg; 200 mg/kg meropenem and 40 mg/kg CMS were given 4 times daily and treatment for all groups continued until 120 h after infection.

### 4.11. Organ Bacterial Load

The organ bacterial eradication of antibiotic treatments was evaluated in same sepsis infection model. The mice were sacrificed, and their liver, kidney and spleen were collected in 2 mL of sterile PBS 18 h post-infection. Thereafter, it was homogenized with a homogenizer. Serial 10-fold dilutions were generated in 1× PBS and plated on meropenem-containing (2 μg/mL) LB agar. Plates were then incubated in 37 °C for 18 h and the number of colonies were counted. Bacterial count was normalized by organ weight. 

### 4.12. Statistical Analysis

Results of the time-kill assays, ROS detection and organ bacterial load experiments were expressed as mean ± standard deviation and evaluated using Student’s *t*-tests. For survival tests, Kaplan–Meier curves were constructed with GraphPad Prism software (v.7.0) and analyzed using Mantel–Cox log-rank tests.

## 5. Conclusions

AS101 displayed strong antimicrobial activity against CRAB. It was able to inhibit bacterial growth at very low concentrations and improved the survival rates of mice through reducing the pathogen levels in the liver, kidney and spleen. Above all, the in vitro and in vivo bactericidal profiles of AS101 indicate that it is an ideal candidate for the urgently needed new antimicrobial agents against carbapenem-resistant *Acinetobacter baumannii*.

## Figures and Tables

**Figure 1 pharmaceuticals-14-00823-f001:**
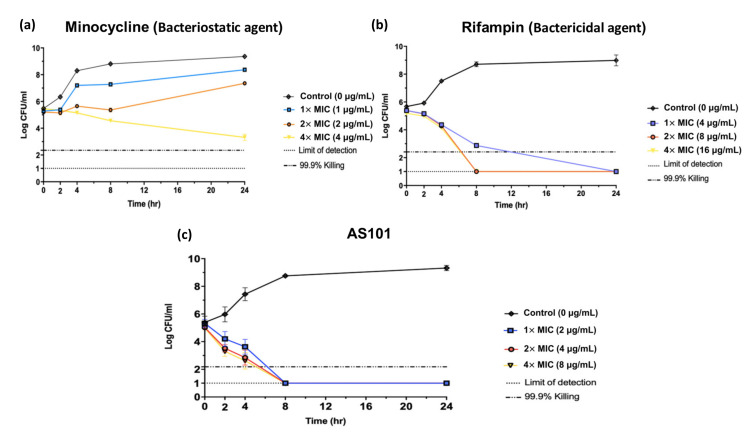
Time kill curve for *A. baumannii* TSP-AB-03 with antibiotics. Bacteria were grown with the indicated concentrations of minocycline (**a**), rifampin (**b**) or AS101 (**c**). Cell densities were measured for three independent cultures by counting the number of colony-forming units of culture for 24 h. Dotted-dashed lines indicate 99.9% reduction in beginning inoculum. Dotted lines indicate detection limits. Error bars are the standard deviation from the mean.

**Figure 2 pharmaceuticals-14-00823-f002:**
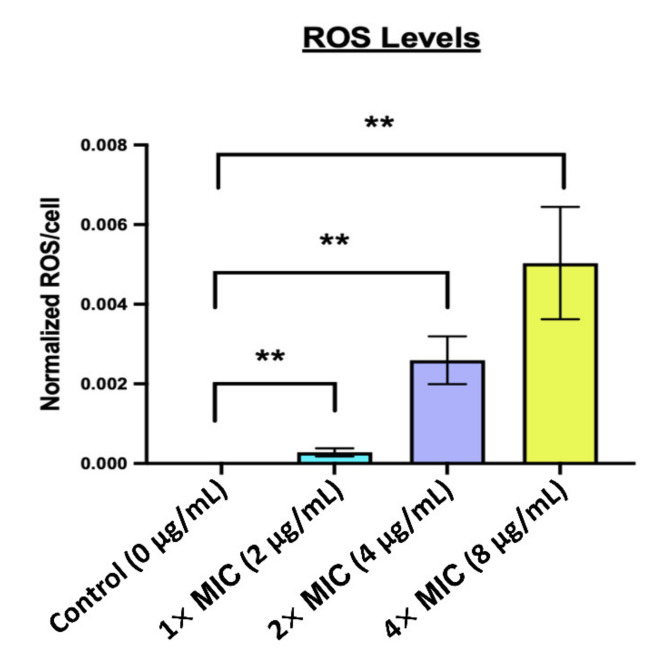
Detection of reactive oxygen species (ROS) generation after the treatment of TSP-AB-03 with AS101. ROS were significantly produced when compared with the control group (0 μg/mL). Data are normalized to viable bacterial counts and expressed as mean ± SD. **, *p* < 0.01.

**Figure 3 pharmaceuticals-14-00823-f003:**
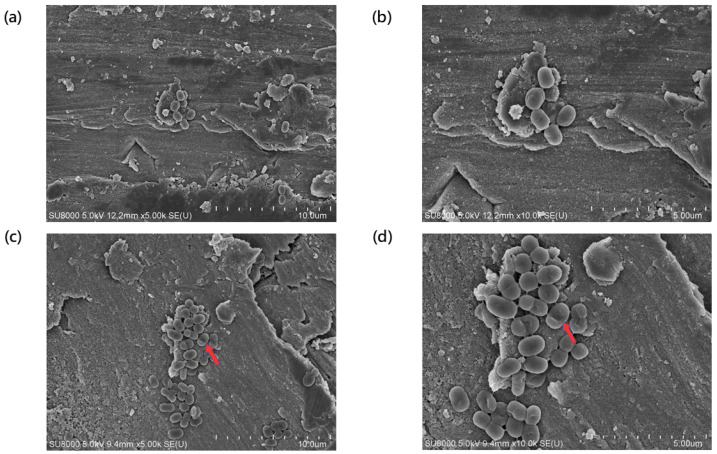
SEM image of TSP-AB-03 strain untreated or treated with AS101. The morphology of untreated group was observed under a magnification of 5000× and a scale bar of 10 μm (**a**) and a magnification of 10,000× and a scale bar of 5 μm (**b**). Those treated with 1× MIC (2 μg/mL) of AS101 were captured under a magnification of 5000× and a scale bar of 10 μm (**c**) and a magnification of 10,000× and a scale bar of 5 μm (**d**). The collapsed appearance (red arrows in (**c**) and (**d**)) indicated the AS101 might attack the bacterial outer membrane.

**Figure 4 pharmaceuticals-14-00823-f004:**
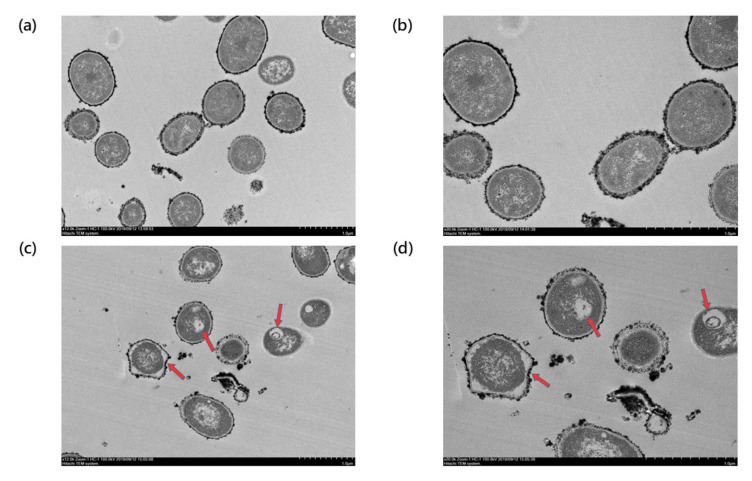
TEM image of TSP-AB-03 strain untreated or treated with AS101. The morphology of untreated group was observed under a magnification of 12,000× and a scale bar of 1 μm (**a**) and 20,000× and a scale bar of 1 μm (**b**). Those treated with 1× MIC (2 μg/mL) of AS101 were captured under a magnification of 12,000× and a scale bar of 1 μm (**c**) and a magnification of 20,000× and a scale bar of 1 μm (**d**). The irregular peri-sphere and vacuole inside bacterial cell (red arrows in (**c**) and (**d**)) implied the bacterial cell lysis.

**Figure 5 pharmaceuticals-14-00823-f005:**
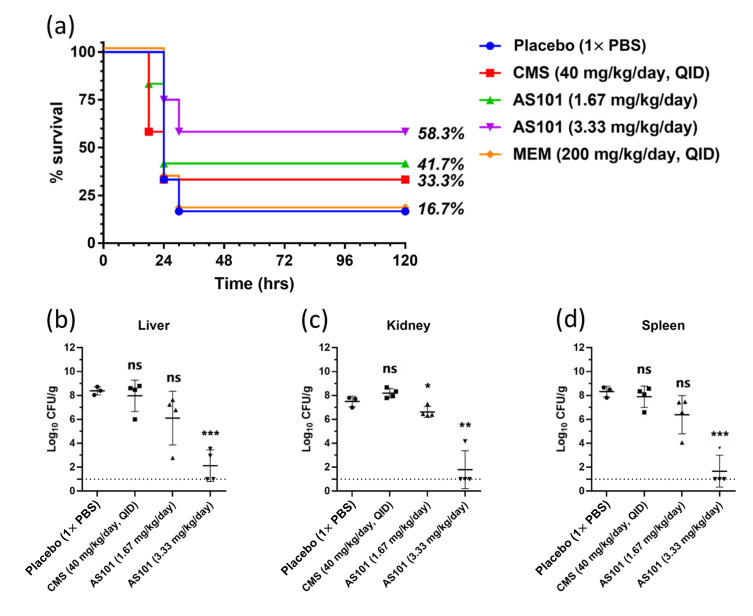
Mice infected with TSP-AB-03 show improved survival rate after treatment with AS101. (**a**), CD-1 mice were intraperitoneally injected with lethal challenges of TSP-AB-03 and treated with 1× PBS (vehicle), meropenem (MEM, 200 mg/kg/day), colistin methanesulfonate (CMS, 40 mg/kg/day) or AS101 at various dosages. Infected mouse viabilities were observed as 16.7% (1/6) for the vehicle treatment; 16.7% (1/6) for MEM; 33.3% (4/12) for CMS; 41.7% (5/12) for AS101 (1.67 mg/kg/day) and 58.3% (7/12) for AS101 (3.33 mg/kg/day). Bacterial clearance was determined using plating counts from the liver (**b**), kidney (**c**) or spleen (**d**) of different treated mice. Placebo: *n* = 3; CMS and AS101: *n* = 4. Data expressed as mean ± SD. ns: no significance; *, *p* < 0.05; **, *p* < 0.01; ***, *p* < 0.001.

**Table 1 pharmaceuticals-14-00823-t001:** Characterization of 27 carbapenem-resistant *A. baumannii* (CRAB) clinical isolates used in this study.

Bacterial Strains	MIC of AS101	Susceptibility to Antimicrobial Agents ^1^							Gene Detection of Carbapenemase
		CL	GM	DOX	CAZ	TIC	LEV	MEM	
TSP-AB-1	8	S	R	R	R	R	R	R	*bla*_OXA-24_, *bla*_OXA-69_
TSP-AB-2	2	S	R	S	R	R	R	R	*bla*_IMP-1_, *bla*_OXA-69_
TSP-AB-3	2	S	R	R	R	R	R	R	*bla*_OXA-24_, *bla*_OXA-69_
TSP-AB-4	8	S	R	R	R	R	R	R	*bla*_OXA-24_, *bla*_OXA-69_
TSP-AB-5	2	S	R	R	R	R	R	R	*bla* _OXA-69_
TSP-AB-6	8	S	R	R	R	R	R	R	*bla* _OXA-69_
TSP-AB-7	2	S	S	S	R	R	I	R	*bla*_OXA-24_, *bla*_OXA-69_
TSP-AB-8	8	S	R	R	R	R	R	R	*bla* _OXA-69_
TSP-AB-9	2	S	R	R	R	R	R	R	*bla*_OXA-24_, *bla*_OXA-69_
TSP-AB-10	1	S	R	R	R	R	R	R	*bla* _OXA-69_
TSP-AB-11	1	S	R	R	R	R	R	R	*bla* _OXA-69_
TSP-AB-12	16	S	R	R	R	R	R	R	*bla* _OXA-69_
TSP-AB-13	32	S	R	R	R	R	R	R	*bla*_VIM-2_, *bla*_OXA-69_
TSP-AB-14	0.5	S	R	R	R	R	R	R	-
TSP-AB-15	4	S	R	R	R	R	R	R	*bla* _OXA-69_
TSP-AB-16	4	S	S	R	R	R	R	R	*bla*_OXA-24_, *bla*_OXA-69_
TSP-AB-17	2	S	R	R	R	R	R	R	-
TSP-AB-18	0.5	S	R	R	R	R	R	R	*bla* _OXA-69_
TSP-AB-19	1	S	R	R	R	R	R	R	-
TSP-AB-20	1	S	R	R	R	R	R	R	-
TSP-AB-21	2	S	I	R	R	R	R	R	*bla*_OXA-24_, *bla*_OXA-69_
TSP-AB-22	2	S	R	R	R	R	R	R	*bla*_OXA-24_, *bla*_OXA-69_
TSP-AB-23	1	S	R	R	R	R	R	R	*bla* _OXA-69_
TSP-AB-24	8	S	I	R	R	R	R	R	*bla* _OXA-69_
TSP-AB-25	1	S	R	R	R	R	R	R	*bla* _OXA-69_
TSP-AB-26	2	S	R	R	R	R	R	R	*bla* _OXA-69_
TSP-AB-27	8	S	S	R	R	R	R	R	*bla* _OXA-69_
Resistant rate	N/A ^2^	0%	81.5%	92.6%	100%	100%	96.3%	100%	N/A
		0/27	22/27	25/27	27/27	27/27	26/27	27/27	

^1^ Abbreviations: CL, Colistin; GM, Gentamicin; DOX, Doxycycline; CAZ, Ceftazidime; TIC, Ticarcillin; LEV, Levofloxacin; MEM, Meropenem. Interpretation of antimicrobial susceptibility was according to criteria recommended by CLSI: S, susceptible; I, intermediate; R, resistant. ^2^ N/A, not applied.

**Table 2 pharmaceuticals-14-00823-t002:** MIC values of AS101 against *A. baumannii* TSP-AB-03 with the pharmacological manipulations.

Treatment	Function	MIC of AS101 Mean ± SD (μg/mL) ^1^
Untreated	-	2 ± 0
^1^ mM EDTA	Alteration of outer-membrane permeability	0.21 ± 0.07
10 mM Ca^2+^	Alteration of outer-membrane charge	4 ± 0
10 mM Mg^2+^	Alteration of outer-membrane charge	4 ± 0
500 mM Mannitol	Reactive oxygen species (ROS) scavenger	24 ± 9.24

^1^ Results show mean and SD of 3 independent replications in each test.

## Data Availability

Data is contained within the article.

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
