# Peer review of "In Vitro and In Vivo Activity of AS101 against Carbapenem-Resistant Acinetobacter baumannii"

_pharmaceuticals, 2021, doi:10.3390/ph14080823_

Round 1

Reviewer 1 Report

The manuscript is well organized, is of great interest, and contains very interesting results. For me, this is ready for publication in its current form.

Author Response

Reviewer 1:

The manuscript is well organized, is of great interest, and contains very interesting results. For me, this is ready for publication in its current form.

Reply: We appreciated reviewer’s comment which really encourage us to make more contributions to this issue. We will check out other reviewers’ suggestions and revise the manuscript.

Reviewer 2 Report

I have reviewed with a lot of interest this manuscript that describes in vitro and in vivo activity of AS101 against carbapenem-resistant Acinetobacter baumannii (CRAB). The methods are solid. The performed experiments are clearly explained. The results and conclusions do match. I have only minor comments regarding this manuscript.

  • AS101 was previously evaluated against multiple bacteria? This needs to be summarized in the introduction or discussion sections. See the following papers among others. https://academic.oup.com/jac/article/67/9/2165/876106; https://link.springer.com/article/10.1007/s00203-009-0490-y
  • Is it possible that immune modulating activity of AS101 improves mice survival rather than bactericidal activity? See https://www.jimmunol.org/content/169/1/384.long
  • Define ROS in abstract and figure 2.
  • Conclusions in abstract and manuscript: Revise to say candidate for further testing in future studies.
  • Introduction: Replace outdated term “septicemia” with either sepsis or bloodstream infection.
  • The authors state that repurposing existing agents is riskless. I disagree. The time and money spent on randomized clinical trials to get approval of these agents in human disease will cost a lot of money and time.
  • Results (lines 90-91): The numbers of carbapenemases do not add up. Please clarify.
  • Spell out CRAB in table 1.
  • Define SEM and TEM prior to first use.
  • What do arrows indicate in figures 3c and 3d?
  • Define CMS prior to first use in manuscript.
  • Figure 5 title: Revise to “treatment with AS101”.
  • Discussion: The notion that carbapenem resistance exceeds 90% is incorrect based on the studies cited in the introduction. Please revise.
  • Many paragraphs in the discussion section (lines 210-213 and 219-224) are redundant as they were previously presented in the introduction. I advise the authors to strengthen the discussion section and avoid redundancy using the discussion points above in comments 1 & 2.

Author Response

Reviewer 2:

I have reviewed with a lot of interest this manuscript that describes in vitro and in vivo activity of AS101 against carbapenem-resistant Acinetobacter baumannii (CRAB). The methods are solid. The performed experiments are clearly explained. The results and conclusions do match. I have only minor comments regarding this manuscript.

Reply: Many thanks to reviewer’s comment. We are glad that our study could be recognized. We revised and reply all the minor comment below. Please check.

AS101 was previously evaluated against multiple bacteria? This needs to be summarized in the introduction or discussion sections. See the following papers among others. https://academic.oup.com/jac/article/67/9/2165/876106; https://link.springer.com/article/10.1007/s00203-009-0490-y

Reply: We appreciated the reviewer’s suggestion. We summarize some available studies of AS101 in the introduction section (line 84-90): “Research in various fields has found that AS101 has a good effect on a variety of diseases, including clinical conditions involving immunosuppression, malignant tumors and AIDS, autoimmune diseases, antiviral, antiparasitic and other diseases [13,15-17]. Some unexpected antibacterial effects of AS101 were also revealed against ESBL-producing Klebsiella pneumoniae and Enterobacter cloacae [18,19]. Recently, another previous examination of AS101 against colistin and carbapenem-resistant K. pneumoniae also documented the antibacterial potential of AS101 (ref).”

Is it possible that immune modulating activity of AS101 improves mice survival rather than bactericidal activity? See https://www.jimmunol.org/content/169/1/384.long

Reply: It’s a really good question for reviewer 2. We added the description about the indirect effect of AS101 to rescue CLP mice model in the introduction section (line 76-78): “An immune-directed treatment effect of AS101 was also observed in a cecal liga-tion-and-punctured mouse model in a previous study [12].”     In our study, we examine the time-kill assay (in vitro) and bacterial organ load (in vivo) to validate the bactericidal activity of AS101 against CRAB.

Define ROS in abstract and figure 2.

Reply: Many thanks to reviewer’s kind reminder. The full meaning of ROS was added (line 148)

Conclusions in abstract and manuscript: Revise to say candidate for further testing in future studies.

Reply: We appreciated the reviewer’s suggestion. The description was revised. (line 42)

Introduction: Replace outdated term “septicemia” with either sepsis or bloodstream infection.

Reply: We appreciated the reviewer’s suggestion. The description was revised. (line 47)

The authors state that repurposing existing agents is riskless. I disagree. The time and money spent on randomized clinical trials to get approval of these agents in human disease will cost a lot of money and time.

Reply: Many thanks to reviewer’s comment. We added a phrase to make the sentence more specifically (line 64-66): “Thus, drug repurposing is considered a much more efficient approach as it is economical and riskless in the drug discovery phase.”

Results (lines 90-91): The numbers of carbapenemases do not add up. Please clarify.

Reply: We are grateful for reviewer’s question. We rechecked the description and the result of gene detection in Table 1. The numbers seem to not add up is owing that some of isolates co-carried more than one carbapenemase.

Spell out CRAB in table 1.

Reply: Many thanks to reviewer’s kind reminder. The full meaning of CRAB was added (line 106)

Define SEM and TEM prior to first use.

Reply: Many thanks to reviewer’s kind reminder. The full meanings of SEM and TEM were added (line 154 and 158-159)

What do arrows indicate in figures 3c and 3d?

Reply: We appreciated reviewer’s kind reminder. We added the description about the images indicated by red arrows in figures 3c and 3d (line 169-170): “The collapsed appearance (red arrows in (c) and (d)) indicated the AS101 might attack the bacterial outer membrane.”

Define CMS prior to first use in manuscript.

Reply: Many thanks to reviewer’s kind reminder. The full meaning of CMS was added (line 184-185)

Figure 5 title: Revise to “treatment with AS101”.

Reply: We appreciated the reviewer’s suggestion. The description was revised. (line 202)

Discussion: The notion that carbapenem resistance exceeds 90% is incorrect based on the studies cited in the introduction. Please revise.

Reply: We apologized for the misleading description. We revised the sentence (line 213): “Carbapenem resistance rate increased yearly and mortality rates for the most common infections of CRAB approach 60%.”

Many paragraphs in the discussion section (lines 210-213 and 219-224) are redundant as they were previously presented in the introduction. I advise the authors to strengthen the discussion section and avoid redundancy using the discussion points above in comments 1 & 2.

Reply: We appreciated the reviewer’s suggestion. We deleted some of descriptions (line 223-227) and summarize AS101 information in the introduction section (line 84-87 and 233-238).

Reviewer 3 Report

This manuscript focuses on the use of a an immunomodulation agent (AS101) as an antibacterial agent against carbapenem-resistant A. baumannii. The subject is interesting because the number of antibiotic-resistant bacterial strains is constantly growing. Although the number of strains tested (only 27) is quite small, in general this study is very complex and well done.

Specific comments are provided below:

  • In the Material and Methods section, points 4.2 and 4.4. are too briefly presented
  • Where were all analyzes performed?

The abbreviations should be explained when they first appear in manuscript

Author Response

Reviewer 3:

This manuscript focuses on the use of a an immunomodulation agent (AS101) as an antibacterial agent against carbapenem-resistant A. baumannii. The subject is interesting because the number of antibiotic-resistant bacterial strains is constantly growing. Although the number of strains tested (only 27) is quite small, in general this study is very complex and well done.

Reply: Many thanks to reviewer’s comment. It is an urgent issue worldwide to figure out solutions against the carbapenem-resistant crisis. Although the strain number in our study is relatively small, the potential of AS101 against CRAB could still be seen. Further testing would be needed for the development of AS101.

Specific comments are provided below:

In the Material and Methods section, points 4.2 and 4.4. are too briefly presented

Reply: We appreciated the reviewer’s suggestion. We revised and added information about result validation and interpretation. “The minimum inhibitory concentration (MIC) of antibiotics were determined by standard agar dilution method. Antibiotics included meropenem, colistin, gentamicin, doxycycline, ticarcillin, levofloxacin and ceftazidime. Two quality-control strains, Escherichia coli ATCC 25922 and Pseudomonas aeruginosa ATCC 27853, were employed to validate the result of each testing. The interpretations of susceptibility results were according to guidelines established by the Clinical and Laboratory Standards Institute (CLSI) [36].” (4.2, line 289-296); “Plasmid DNA was prepared using Presto™ Mini Plasmid Kit (PDH300) (Geneaid, Taipei, Taiwan). Genomic DNA was purified manually as described in a previous study [37]. PCR methods were used to detect carbapenemase including blaKPC, blaNDM, blaIMP-1, blaIMP-2, blaNMC, blaSME, blaVIM-1, blaVIM-2, blaSPM-1, blaGIM-1, blaSIM-1, blaIMI, blaGES, blaOXA-24, blaOXA-48, and blaOXA-69, respectively [38,39]. The positive PCR products were subject to sequence for validation of the results. Positive controls were also included in each reaction to rule out operation error.” (4.4, line 306-312)

Where were all analyzes performed?

Reply: We are grateful for the reviewer’s question. Dr. Hung’s lab, a biosafety level (BSL) 2 lab, is an academic research laboratory in Kaohsiung Medical University (KMU). The lab safety checks for laboratories were conducted annually by the KMU office of environmental health and safety has many in KMU. The animal study was conducted in the KMU Laboratory Animal Center, an Association for Assessment and Accreditation of Laboratory Animal Care International (AAALAC)-accredited facility. The description about AAALAC was added in the Materials and methods section. (4.9, line 350-352)

The abbreviations should be explained when they first appear in manuscript.

Reply: We appreciated the reviewer’s kind reminder. We checked the whole article and added the full meaning of each abbreviation, including ROS, SEM, TEM, and so on.